# REFLECTION FROM RETRIEVAL: MLLM-GUIDED ITERATIVE REASONING FOR ZERO-SHOT COMPOSED IMAGE RETRIEVAL

## ABSTRACT

Zero-Shot Composed Image Retrieval (ZS-CIR) aims to retrieve a target image based on a reference image and a modification text, without requiring task-specific training. Most existing methods directly rewrite the query from the multimodal inputs without verification or self-correction, making initial misinterpretations of the user's intent unrecoverable and leading to retrieval failure. To address this limitation, we propose CoRR, a novel training-free framework that reframes ZS-CIR as a dynamic and self-correcting process. In contrast to prior methods, CoRR incorporates evidence from retrieved results as explicit feedback and employs a Multimodal Large Language Model (MLLM) to iteratively refine query representations through a Chain-of-Thought reasoning process. In order to ensure stable query evolution, we employ Spherical Linear Interpolation (Slerp) to fuse historical and newly generated query. Furthermore, we introduce Retrieval-Driven Caption Optimization, which supplies the MLLM with high-fidelity contextual examples to enhance its reasoning and ensure that outputs align with the preferences of the embedding space. Extensive experiments on multiple benchmarks, including CIRCO, CIRR, and FashionIQ, demonstrate that CoRR significantly outperforms existing state-of-the-art methods, establishing the superior effectiveness of our proposed paradigm.

## 1 INTRODUCTION

Composed Image Retrieval (CIR) aims to retrieve a target image that preserves the relevant visual content of a reference image while incorporating the semantic modifications described in a textual query (Vo et al., 2019; Delmas et al., 2022; Huynh et al., 2025; Xing et al., 2025). This fine-grained retrieval task has significant practical value in numerous real-world scenarios like web search and e-commerce (Chen et al., 2020; Saito et al., 2023; Bai et al., 2024; Tang et al., 2025b), offering users a more intuitive and flexible way to interact with visual content.

Zero-Shot Composed Image Retrieval (ZS-CIR) (Saito et al., 2023; Karthik et al., 2024; Yang et al., 2024; Tang et al., 2025a;b; Luo et al., 2025) has emerged as a promising paradigm due to its cost-effectiveness. A common practice is to first employ Multimodal Large Language Models (MLLMs) or Large Language Models (LLMs) to generate composed queries that integrate information from both the reference image and modification text, then to utilize pre-trained multi-modal embedding models like CLIP (Radford et al., 2021) to retrieve results from the target database.

The effectiveness of these methods heavily relies on the accuracy of the composed queries, which require precise semantic editing of the reference image based on the provided text. However, this task presents a challenge due to its inherent ambiguity and complexity: the semantics of the target image are not strictly determined by the reference image and the modification text (Bordogna & Pasi, 1993; Chen & Wang, 2002; Yang et al., 2024). As a result, the generated text queries may either become overly broad, potentially omitting critical visual elements, or excessively specific, emphasizing irrelevant details from the reference image, both of which will lead to erroneous results.

Recent efforts have leveraged the reasoning capabilities of MLLMs and LLMs to better interpret user intent. However, the focus of such approaches is restricted exclusively to query analysis and rewriting, as illustrated in Figure 1 (a). For instance, OsrCIR (Tang et al., 2025b) utilizes an MLLM to

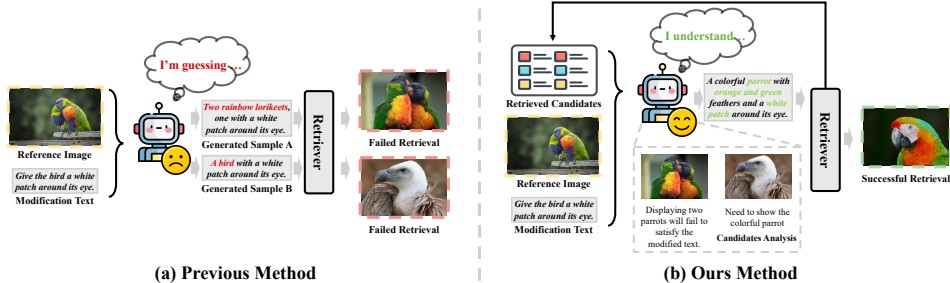

Figure 1: An overview of the previous methods and our method. (a) relies solely on guessing the user's intent based on the query itself, while (b) analyzes and improves the query through feedback obtained from retrieval results.

analyze reference images and modification text, employing Chain of Thought (CoT) to reason about explicit intentions of the user. Similarly, LDRE (Yang et al., 2024) identifies the inherent fuzziness of this task and seeks to generate diverse queries from multiple perspectives, thereby encompassing the full range of potential semantics of the target composition. CoTMR (Sun et al., 2025) further enhances this process by integrating multi-scale reasoning with CoT, enabling comprehensive inference through fine-grained predictions regarding the presence or absence of key elements at the object level. While these approaches optimize queries by improving the alignment between visual and textual inputs, there exists another line of work that focuses on utilizing pseudo-relevance feedback to expand queries (Cao et al., 2008; Wang et al., 2021). These methods primarily collect Top-K retrieval results and adjust queries based on overall shifts in feature spaces. However, this approach may overlook the noise present in the retrieved results, potentially leading to erroneous outcomes.

In this paper, we propose CoRR (**C**hain **o**f **R**eflective Composed Image **R**etrieval), a novel training-free framework that effectively integrates multimodal inputs and retrieval feedback for query refinement. CoRR facilitates a reflective analysis and enhances the refinement of the composed query not only based on the reference image and text but also on the retrieved images. As shown in Figure 1 (b), we employ an MLLM to assess the evidence within the retrieved images, analyzing whether the constraints specified in the original query are satisfied. This analysis aids in identifying which elements should be preserved and which should be modified, ultimately leading to a more accurate query generation. However, the direct application of such reflective chains can be problematic, as we often encounter "query drift" (Mitra et al., 1998) during multiple refinement iterations. To mitigate this issue, we propose a Historical Query Fusion strategy, which employs Spherical Linear Interpolation (Slerp) (Shoemake, 1985) to seamlessly integrate historical query vectors, thereby ensuring a stable and progressive reflective refinement.

To further enhance the performance, we identify that an effective query must not only accurately represent the user's intent but also align well with the embedding space of the retrieval model. To achieve better alignment, we propose integrating retrieval-optimized captions for each image into the reflection process. These captions are produced through a self-retrieval process within the database, which highlights each image's distinctive visual characteristics, thereby enhancing its discriminative representation. By incorporating these optimized captions into our reflection framework, we can not only accurately identify key evidence for each image and propagate that information to refine the query, but also emulate the styles and patterns to generate query that is more conducive to retrieval.

Our main contributions are as follows:

- We introduce CoRR, a novel training-free framework that reframes ZS-CIR as a dynamic and self-correcting process. CoRR is plug-and-play and compatible with existing methods, thereby consistently enhancing overall performance.
- We introduce an MLLM-guided self-reflection mechanism that reasons about evidence from both multimodal inputs and retrieved feedback, combined with Slerp-based historical query fusion. It can achieve stable and progressive query refinement that better captures the user's intent.
- We propose generating retrieval-optimized captions and incorporating them into self-reflection, allowing queries to align more effectively with the employed retrieval models.
- Through extensive experiments on standard benchmarks including FashionIQ, CIRR, and CIRCO, we demonstrate that CoRR achieves state-of-the-art performance, improving existing methods by 3 to 9 percentage points in a variety of models with only two additional rounds of self-reflection, highlighting the effectiveness of our iterative retrieval paradigm.

## 2 RELATED WORK

**Composed Image Retrieval.** Composed Image Retrieval (CIR) enables the retrieval of target images by leveraging a reference image in conjunction with textual modification texts (Wu et al., 2021; Han et al., 2017; Vo et al., 2019). Classical approaches to CIR typically relied on specialized models trained on large-scale, manually annotated triplet datasets (Liu et al., 2021; Baldrati et al., 2022; Chen et al., 2020; Chen & Bazzani, 2020; Lee et al., 2021; Anwaar et al., 2021). The core technique involved projecting visual and textual features into a shared embedding space, often using contrastive learning objectives (Sohn, 2016; Radford et al., 2021; Roth et al., 2022). To avoid reliance on annotated triplets, recent work has turned to zero-shot CIR, which performs retrieval without task-specific training data. One strategy leverages pseudo-tokens (Saito et al., 2023; Baldrati et al., 2023; 2022; Gu et al., 2024; Tang et al., 2025a; bai et al., 2024) to encode reference images, subsequently combining them with the reference caption. However, these techniques typically encounter difficulties in understanding implicit human intentions embedded in manipulation text. Recent advances in Multimodal Large Language Models (MLLMs) have catalyzed the emergence of novel methodologies that transform CIR tasks into text-to-image retrieval paradigms (Karthik et al., 2024; Yang et al., 2024; Tang et al., 2025b). This paradigm shift leverages MLLMs to convert reference images and modification texts into descriptive captions, enabling text-to-image retrieval using models like CLIP (Radford et al., 2021). However, these approaches only focus on analyzing the query based on the reference image and modification text, lacking a mechanism to verify whether the refined query aligns with the actual visual evidence, leading to misalignment with user intent and limits retrieval accuracy.

**Embedding Models and Multimodal Large Language Models.** Embedding models, particularly pioneering architectures such as CLIP (Radford et al., 2021) and BLIP (Li et al., 2022), have successfully established a unified semantic space by mapping images and text through training on massive image-text datasets. This breakthrough has enabled diverse applications across image generation (Kim et al., 2022; Rombach et al., 2022), classification (Zhou et al., 2022; Qu et al., 2025), and cross-modal retrieval (Bogolin et al., 2022; Wang et al., 2025). Recently, the field has evolved from simple feature alignment toward deep integration of visual capabilities with Large Language Models (LLMs), resulting in more powerful Multimodal Large Language Models (MLLMs) (Liu et al., 2024; Zhu et al., 2025; Bai et al., 2025; Shahriar et al., 2024). These models achieve deep visual understanding and complex reasoning capabilities through instruction tuning, enabling tasks such as visual question answering and image description generation. Existing work has demonstrated that combining Embedding Models with MLLMs can efficiently perform Composed Image Retrieval (CIR) tasks in a single inference. Our work extends this powerful "retriever + reasoner" paradigm by introducing an iterative correction mechanism to address its inherent retrieval limitations.

## 3 METHODOLOGY

### 3.1 PRELIMINARIES

Before formally presenting our method, we provide a detailed definition of the ZS-CIR task. Given a reference image $I_r$ and a modification text $T$, the objective of Composed Image Retrieval (CIR) is to identify an image from a database $\mathcal{D} = \{I_1, I_2, ..., I_n\}$, that best represents the reference image $I_r$ after applying the semantic changes described by the modification text $T$.

While traditional supervised methods train a composition function using costly-to-acquire <reference, text, target> triplets, ZS-CIR methods avoid this by converting visual features to textual representations for retrieval. Despite their different strategies, these approaches can be abstracted into a unified formulation. We can consider them as employing a universal multi-modal embedding model $\Psi(*)$, which is capable of processing unimodal inputs (an image or a text) or their composition, projecting them into a shared embedding space. The composed query vector is thus $v_q = \Psi(I_r, T) \in \mathbb{R}^d$, and the candidate image vectors are $\Psi_I(I_i)$. where $d$ is the dimension of the shared embedding space and the $\Psi_I$ denotes an image-only input. The retrieval goal is to find the image $I^*$ from the candidates $\mathcal{D}$ that maximizes their similarity:

$$I^* = \Theta(\mathcal{D}, I_r, T) = \arg\max_{I_i \in \mathcal{D}} \frac{\Psi_I(I_i)^\top \Psi(I_r, T)}{\|\Psi_I(I_i)\| \cdot \|\Psi(I_r, T)\|} \tag{1}$$

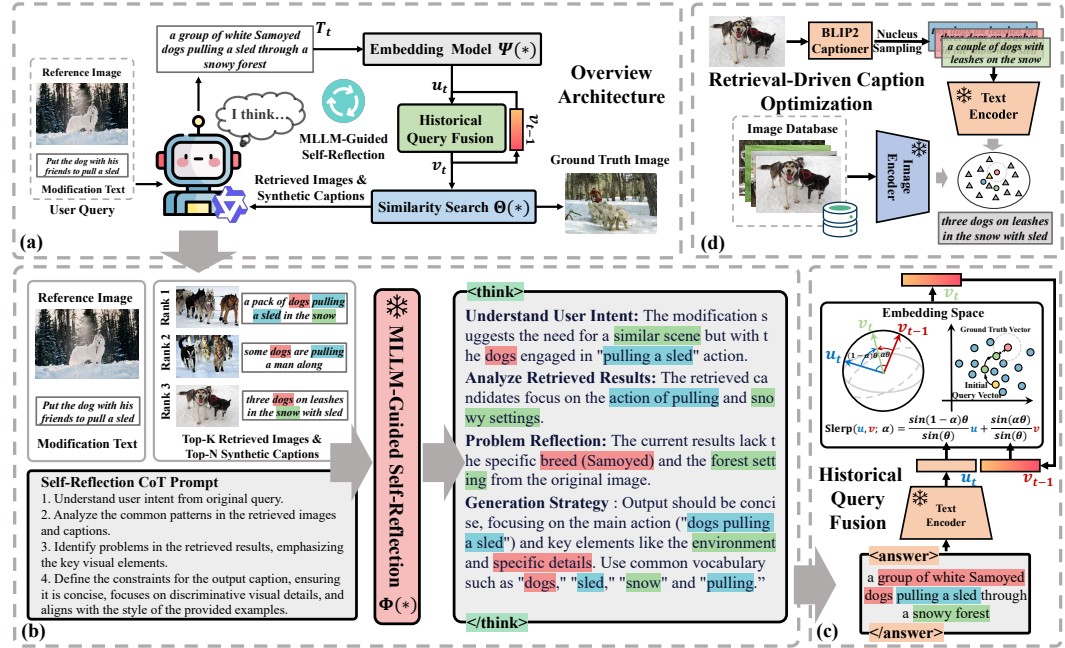

Figure 2: An architecture of our framework. (a) illustrates the simplified pipeline of our proposed method. (b) demonstrates the reasoning and self-reflection process facilitated by the MLLM, where different colors highlight distinct key visual elements. (c) visualizes the Historical Query Fusion process, while (d) showcases our Retrieval-Driven Caption Optimization strategy.

## 3.2 OVERVIEW

The pipline of our proposed framework is illustrated in Figure 2 (a). Our proposed CoRR framework follows "retrieval-reflection-refinement" iterative process. It begins by generating an initial query vector $v_0 = \Psi(I_r, T)$ from reference image $I_r$ and modification text $T$ using the embedding model to retrieve a set of candidate images $\mathcal{I}_0$. Subsequently, in $t$ round, an MLLM reasoner $\Phi$ analyzes the Top-K (default 5) retrieved images $\mathcal{I}_{t-1}$ from the previous retrieval and their associated Top-N (default 10) synthetic captions $\mathcal{C}_{t-1}$, along with the original inputs $(I_r, T)$, to generate a new textual query $T'_t$. This refined text query is then used to compute an updated query vector $v_t$ via Historical Query Fusion for next round.

Our approach primarily consists of three core modules: (1) MLLM-Guided Self-Reflection, (2) Historical Query Fusion based on Slerp, and (3) Retrieval-Driven Caption Optimization. We will elaborate on each in detail.

## 3.3 MLLM-GUIDED SELF-REFLECTION

In each retrieval loop, our MLLM Reasoner executes a CoT process to analyze the retrieved candidates from the previous round, reflect on the results, and generate a refined query for the next iteration. Based on the examples given in the Figure 2 (b), we briefly introduce the process. For specific CoT prompts, please refer to the appendix A.1.

**Understand User Intent.** First, the MLLM will analyze the reference image and modification text to clarify user intent. As shown in Figure, the MLLM finds that user intent is to retrieve an image of dogs pulling a sled in an environment similar to the original scene.

**Analysis Retrieved Results.** Then, the MLLM analyzes the retrieved images and synthetic captions from the previous round. This step aims to retain key visual elements that match user intent. As shown in Figure, the MLLM examines the caption of "pulling" and "snow settings".

**Problem Reflection.** Building upon the analysis of retrieved results, the MLLM synthesizes its findings to identify problem between current results and user intent. As shown in Figure, the MLLM finds that the current results lack the specific breed and forest environment.

**Generation Strategy.** In the final step, the MLLM will be guided by two key criteria: (1) accurate integration of key visuals, incorporating corrective insights from problem reflection and (2) constraining the final output by learning from synthetic captions to align with the embedding model's preferences. The MLLM clarifies that the output needs to focus on the action, the environment, and the missing breed. It also specifies that the output should be concise and use common vocabulary.

## 3.4 HISTORICAL QUERY FUSION BASED ON SLERP

As shown in Figure 2 (c), in each iteration of our framework, the MLLM generates a refined target caption $T_t$, which is then encoded into a new query vector $u_t = \Psi_T(T_t)$. However, directly replacing the previous query vector with this new one can lead to instability. The MLLM's reflection might cause an over-correction or introduce noise, leading to "query drift" (Mitra et al., 1998) where valuable information from the previous state is lost.

To ensure a smooth and stable evolution of the query, we fuse the historical query $v_{t-1}$ from the previous iteration with the new query vector $u_t$. We employ Spherical Linear Interpolation (Slerp) (Shoemake, 1985) for this task, as it is naturally suited for operating on the hypersphere of unit-normalized embedding models. Slerp ensures a smooth transition between the two vectors, providing a robust update mechanism. We prove its validity in Section A.4. The final query vector for the current iteration $v_t$ is computed as follows:

$$v_t \longleftarrow \text{Slerp}(u_t, v_{t-1}; \alpha) \tag{2}$$

The Slerp function is defined as:

$$\text{Slerp}(u, v; \alpha) = \frac{\sin(1-\alpha)\theta}{\sin(\theta)}u + \frac{\sin(\alpha\theta)}{sin(\theta)}v \tag{3}$$

where $\theta = \arccos(u \cdot v)$ is the angle between the two vectors, and $\alpha \in [0, 1]$ (default 0.8) is a fixed hyperparameter that controls the interpolation weight, balancing the influence of the historical query and the new evidence. See Section 4.3 for ablation studies. This Slerp-based fusion mitigates destructive drift by maintaining momentum from the previous query while still integrating the new insights from the MLLM's reflection. It effectively smooths the search trajectory within the embedding space as shown in Figure 2 (c). The resulting updated vector $v_t$, is then used to retrieve the image gallery $\mathcal{D}$ with $\Theta(*)$ to produce the candidate set for the next iteration.

## 3.5 RETRIEVAL-DRIVEN CAPTION OPTIMIZATION

Many studies have shown that dense embedding models are highly sensitive to their input (Rafiei Asl et al., 2024; Magomere et al., 2025). To align the MLLM outputs with the preferences of the embedding model, prior methods have attempted to guide generation through fixed and manually-curated image caption examples (Karthik et al., 2024; Tang et al., 2025b). However, these static examples are inherently disconnected from the current query and fail to provide relevant, adaptive guidance.

Our key insight is that the effective examples for guiding the MLLM should be dynamic and inherently query-relevant. By providing these relevant and retrieval-focused captions to the MLLM, we guide it to learn the embedding model's preferences, including semantic granularity and sentence style. Additionally, providing extra captions allows the MLLM to learn the discriminative visual elements within the images, which helps it better analyze the retrieved results.

Therefore, As shown in Figure 2 (d), our Retrieval-Driven Caption Optimization strategy generates $M$ (default 30) high-quality captions $\{C_i\}_{i=1}^{M}$ for each of the top-N retrieved images using BLIP2 (Li et al., 2023b). To select the most effective one caption, we employ a retrieval-based validation approach that ranks captions using a two-stage sorting strategy: first by the rank of the source image $I$ when using caption $C_i$ as a query, then by the similarity score between the caption and the source image for captions with identical ranks. Formally, for each caption $C_i$, we compute its ranking:

$$(r_i, s_i) < (r_j, s_j) \iff (r_i < r_j) \text{ or } (r_i = r_j \text{ and } s_i > s_j) \tag{4}$$

where $r_i = \text{rank}(I, C_i)$ denotes the rank of image $I$ when caption $C_i$ is used as a query, and $s_i = \text{sim}(I, C_i)$ represents the similarity score between caption $C_i$ and image $I$. We provide corresponding ablation studies in Section 4.3 and present visual analysis in Section A.3.

## 4 EXPERIMENTS

**Implementation Details.** Our framework is implemented in PyTorch(Paszke et al., 2019) and all experiments are conducted on a single NVIDIA A6000 GPU with 48GB. For image retrieval, we use a FAISS (Douze et al., 2024) Flat index to perform an exact search, using inner product as the similarity metric. We adopt different existing ZS-CIR models as the embedding models. In init round, retrieval is performed solely by the embedding model. Subsequently, the MLLM executes two additional reflection and refinement rounds to iteratively update the query. Our primary MLLM is Qwen-VL-Max [1]. we leverage BLIP-2 (Li et al., 2023a) with a OPT-6.7b (Zhang et al., 2022) language model as the image captioner to generate optimized caption examples.

**Datasets and Evaluation Metrics.** We evaluate our method's performance on three widely-recognized benchmarks (**CIRR** (Liu et al., 2021), **CIRCO** (Baldrati et al., 2023) and **FashionIQ** (Wu et al., 2021)) for composed image retrieval. Moreover, the experimental results of **GeneCIS** (Vaze et al., 2023) are included in the appendix A.2 due to space limitations. CIRCO and CIRR are designed for object modification tasks, where reference images provide guidance for altering objects or backgrounds. Following the official protocols, we evaluate CIRCO using mean Average Precision at k (mAP@k) and CIRR using recall at k (Recall@k), both via their respective evaluation servers on the hidden test sets. Additionally, for CIRR, we report subset recall (Recall$_{sub}$@k), which measures the ability to identify the target within a restricted set of relevant images. In contrast, FashionIQ focuses on attribute adjustment, leveraging textual descriptions to change specific image attributes. We report recall at k (Recall@k) on the validation set for each category (Shirt, Dress and Toptee).

**Baseline and Backbone.** To evaluate the effectiveness of our module, we conduct comprehensive comparisons against several prior state-of-the-art ZS-CIR models. All baselines are built upon pre-trained CLIP weights (Radford et al., 2021), allowing for fair comparisons within the same parameter scale. Specifically, we group our comparisons by the underlying CLIP architecture: **CLIP-ViT-B/32**, **CLIP-ViT-L/14**. Since **MMRet-Base** (Zhou et al., 2025) is only available for the **CLIP-ViT-B/16** variant, our comparison with it is limited to this version. To validate our approach, we benchmark it against a variety of leading training-free and training-based Composed Image Retrieval methods. Our comparison set includes **SEARLE** (Baldrati et al., 2023), **Pic2Word** (Saito et al., 2023), **Slerp-TAT** (Jang et al., 2024), **LDRE** (Yang et al., 2024), **Context-I2W** (Tang et al., 2024), **CIReVL** (Karthik et al., 2024), **ImageScope** (Luo et al., 2025), **PrediCIR** (Tang et al., 2025a), **OSrCIR** (Tang et al., 2025b) and **MMRet** (Zhou et al., 2025). By integrating our module with these backbones (e.g., "Ours+MMRet-large"), we achieve substantial performance gains over the original methods, highlighting the consistent benefits of our approach.

### 4.1 MAIN RESULTS

We present our main quantitative results on the CIRCO and CIRR benchmarks in Table 1, and FashionIQ benchmark in Table 2. The results demonstrate that our proposed paradigm consistently and significantly enhances the performance of various baseline methods across different architectures.

In Table 1, we evaluate our approach on the CIRCO and CIRR datasets to demonstrate its performance on tasks that require both foreground-background separation and fine-grained modifications. Specifically, on CIRCO, using the CLIP-ViT-L backbone with Slerp ("Ours+Slerp"), we enhances the mAP@5 score from 16.40% to 26.08%, representing a relative improvement of over 59%. Notably, this is achieved in a training-free manner, significantly outperforming Slerp+TAT (Jang et al., 2024), the original training-based method proposed in its paper. Moreover, when applied to a strong baseline such as MMRet-Large, our method still achieves a consistent uplift, improving mAP@5 from 40.2% to 42.7%. Furthermore, on CIRR, when applied to MMRet-Base, our method increases Recall@1 from 35.97% to 41.58% (+5.61). Similarly, with MMRet-Large, Recall@1 rises from 37.95% to 43.21% (+5.26).

On the domain-specific FashionIQ benchmark, which demands precise localization of specific fashion attributes, our method demonstrates strong effectiveness. As illustrated in Table 2, our approach consistently improves the performance of various baseline models, achieving an average increase of 2-5 points in R@10 and R@50 across all three categories. These results highlight our method's ability to capture fine-grained, domain-specific attributes with notable accuracy.

---

[1] https://github.com/QwenLM/Qwen-VL

Table 1: **Comparison on CIRCO and CIRR Test Data.** Results are grouped by architecture and sorted by publication year. Training-free methods are marked with "✓". Methods with "†" are our implementations. Green highlighting indicates the best performance, while blue highlighting indicates the second-best performance.

| | Architecture | Training-free | CIRCO mAP@k | | | | CIRR Recall@k | | | | CIRR Recall$_{sub}$@k | | |
|---|---|---|---|---|---|---|---|---|---|---|---|---|---|
| | | | k=5 | k=10 | k=25 | k=50 | k=1 | k=5 | k=10 | k=50 | k=1 | k=2 | k=3 |
| CLIP-ViT-B | SEARLE (ICCV'23) | | 9.35 | 9.94 | 11.13 | 11.84 | 24.00 | 53.42 | 66.82 | 89.78 | 54.89 | 76.60 | 88.19 |
| | Slerp (ECCV'24)† | ✓ | 6.51 | 7.05 | 8.13 | 8.70 | 18.12 | 49.11 | 63.16 | 87.59 | 61.99 | 80.61 | 90.94 |
| | **Ours+Slerp** | ✓ | 14.42 | 14.99 | 16.55 | 17.44 | 24.77 | 56.02 | 70.27 | 91.66 | 63.54 | 82.72 | 91.59 |
| | Δ (Ours vs Slerp) | | (+7.91) | (+7.94) | (+8.42) | (+8.74) | (+6.65) | (+6.91) | (+7.11) | (+4.07) | (+1.55) | (+2.11) | (+0.65) |
| | Slerp+TAT (ECCV'24) | | 9.34 | 10.26 | 11.65 | 12.33 | 28.19 | 55.88 | 68.77 | 88.51 | 61.13 | 80.63 | 90.68 |
| | LDRE (SIGIR'24) | ✓ | 17.96 | 18.32 | 20.21 | 21.11 | 25.69 | 55.13 | 69.04 | 89.90 | 60.53 | 80.65 | 90.70 |
| | CIReVL (ICLR'24) | ✓ | 14.94 | 15.42 | 17.00 | 17.82 | 23.94 | 52.51 | 66.0 | 86.95 | 60.17 | 80.05 | 90.19 |
| | ImageScope (WWW'25) | ✓ | 22.36 | 22.19 | 23.03 | 23.83 | 34.36 | 60.58 | 71.40 | 88.41 | 74.63 | 87.93 | 93.83 |
| | OSrCIR (CVPR'25) | ✓ | 18.04 | 19.17 | 20.94 | 21.85 | 25.42 | 54.54 | 68.19 | - | 62.31 | 80.86 | 91.13 |
| | MMRet-Base (ACL'25)† | | 34.21 | 34.78 | 37.20 | 38.38 | 35.97 | 68.17 | 79.56 | 94.72 | 71.61 | 87.47 | 94.46 |
| | **Ours+MMRet-Base** | ✓ | 37.22 | 37.94 | 40.4 | 41.55 | 41.58 | 72.31 | 82.48 | 96.12 | 74.77 | 90.1 | 95.66 |
| | Δ (Ours vs MMRet-Base) | | (+3.01) | (+3.16) | (+3.2) | (+3.17) | (+5.61) | (+4.14) | (+2.92) | (+1.4) | (+3.16) | (+2.63) | (+1.4) |
| CLIP-ViT-L | Pic2Word (CVPR'23) | | 8.72 | 9.51 | 10.64 | 11.29 | 23.90 | 51.70 | 65.30 | 87.80 | - | - | - |
| | SEARLE-XL (ICCV'23) | | 11.68 | 12.73 | 14.33 | 15.12 | 24.24 | 52.48 | 66.29 | 88.84 | 53.76 | 75.01 | 88.19 |
| | Context-I2W (AAAI'24) | | - | - | - | - | 25.60 | 55.10 | 68.50 | 89.80 | - | - | - |
| | LinCIR (CVPR'24) | | 12.59 | 13.58 | 15.00 | 15.85 | 25.04 | 53.25 | 66.68 | - | 57.11 | 77.37 | 88.89 |
| | Slerp (ECCV'24)† | ✓ | 16.40 | 18.41 | 20.89 | 21.97 | 19.28 | 48.22 | 62.24 | 85.74 | 58.05 | 78.05 | 88.96 |
| | **Ours+Slerp** | ✓ | 26.08 | 27.65 | 30.48 | 31.74 | 25.59 | 56.75 | 70.12 | 90.84 | 62.99 | 81.64 | 90.94 |
| | Δ (Ours vs Slerp) | | (+9.68) | (+9.24) | (+9.59) | (+9.77) | (+6.31) | (+8.53) | (+7.88) | (+5.1) | (+4.94) | (+3.59) | (+1.98) |
| | Slerp+TAT (ECCV'24) | | 18.46 | 19.41 | 21.43 | 22.41 | 30.94 | 59.4 | 70.94 | 89.18 | 64.7 | 82.92 | 92.31 |
| | LDRE (SIGIR'24) | ✓ | 23.35 | 24.03 | 26.44 | 27.3 | 26.53 | 55.57 | 67.54 | 88.50 | 60.43 | 80.31 | 89.90 |
| | CIReVL (ICLR'24) | ✓ | 18.57 | 19.01 | 20.89 | 21.8 | 24.55 | 52.31 | 64.92 | 86.34 | 59.54 | 79.88 | 89.69 |
| | ImageScope (WWW'25) | ✓ | 25.39 | 25.82 | 27.07 | 27.98 | 34.99 | 61.35 | 71.49 | 88.84 | 74.94 | 88.24 | 94.0 |
| | PrediCIR (CVPR'25) | | 15.70 | 17.10 | 18.60 | 19.30 | 27.20 | 57.00 | 70.20 | - | - | - | - |
| | OSrCIR (CVPR'25) | ✓ | 23.87 | 25.33 | 27.84 | 28.97 | 29.45 | 57.68 | 69.86 | - | 62.12 | 81.92 | 91.1 |
| | MMRet-Large (ACL'25)† | | 40.20 | 41.20 | 43.80 | 44.91 | 37.95 | 70.36 | 81.08 | 94.75 | 73.23 | 88.12 | 94.8 |
| | **Ours+MMRet-Large** | ✓ | 42.70 | 44.09 | 46.90 | 48.04 | 43.21 | 73.83 | 83.9 | 95.78 | 76.82 | 90.31 | 96.1 |
| | Δ (Ours vs MMRet-Large) | | (+2.5) | (+2.89) | (+3.1) | (+3.13) | (+5.26) | (+3.47) | (+2.82) | (+1.03) | (+3.59) | (+2.19) | (+1.3) |

Table 2: **Comparison on FashionIQ Validation Data.** Results are grouped by architecture and sorted by publication year. Training-free methods are marked with "✓". Methods with "†" are our implementations. Green highlighting indicates the best performance, while blue highlighting indicates the second-best performance.

| | Architecture | Training-free | Shirt | | Dress | | Toptee | | Average | |
|---|---|---|---|---|---|---|---|---|---|---|
| | | | R@10 | R@50 | R@10 | R@50 | R@10 | R@50 | R@10 | R@50 |
| CLIP-ViT-B | SEARLE (ICCV'23) | | 24.44 | 41.61 | 18.54 | 39.51 | 25.70 | 46.46 | 22.89 | 42.53 |
| | Slerp (ECCV'24)† | ✓ | 22.18 | 39.40 | 19.98 | 39.76 | 26.31 | 44.01 | 22.82 | 41.06 |
| | Ours+Slerp | ✓ | 26.94 | 45.44 | 22.16 | 42.89 | 29.78 | 50.28 | 26.29 | 46.20 |
| | Δ (Ours vs Slerp) | | (+4.76) | (+6.04) | (+2.18) | (+3.13) | (+3.47) | (+6.27) | (+3.47) | (+5.14) |
| | Slerp+TAT (ECCV'24) | | 23.06 | 41.95 | 19.24 | 42.14 | 26.57 | 47.78 | 22.96 | 43.96 |
| | LDRE (SIGIR'24) | ✓ | 27.38 | 46.27 | 19.97 | 41.84 | 27.07 | 48.78 | 24.81 | 45.63 |
| | CIReVL (ICLR'24) | ✓ | 28.36 | 47.84 | 25.29 | 46.36 | 31.21 | 53.85 | 28.29 | 49.35 |
| | ImageScope (WWW'25) | ✓ | 24.29 | 37.49 | 18.0 | 35.20 | 24.99 | 41.41 | 22.42 | 38.03 |
| | OSrCIR (CVPR'25) | ✓ | 31.16 | 51.13 | 29.35 | 50.37 | 36.51 | 58.71 | 32.34 | 53.40 |
| | MMRet-Base (ACL'25)† | | 33.81 | 53.14 | 26.28 | 49.38 | 36.1 | 57.32 | 32.06 | 53.28 |
| | Ours+MMRet-Base | ✓ | 36.85 | 55.74 | 27.12 | 50.42 | 38.19 | 58.80 | 34.05 | 54.99 |
| | Δ (Ours vs MMRet-Base) | | (+3.04) | (+2.6) | (+0.84) | (+1.04) | (+2.09) | (+1.48) | (+1.99) | (+1.71) |
| CLIP-ViT-L | Pic2Word (CVPR'23) | | 26.20 | 43.60 | 20.00 | 40.20 | 27.90 | 47.40 | 24.70 | 43.73 |
| | SEARLE-XL (ICCV'23) | | 26.89 | 45.58 | 20.48 | 43.13 | 29.32 | 49.97 | 25.56 | 46.23 |
| | Context-I2W (AAAI'24) | | 29.70 | 48.60 | 23.10 | 45.30 | 30.60 | 52.90 | 27.80 | 48.90 |
| | LinCIR (CVPR'24) | | 29.10 | 46.81 | 20.92 | 42.44 | 28.81 | 50.18 | 26.28 | 46.49 |
| | Slerp (ECCV'24)† | ✓ | 27.58 | 42.89 | 21.42 | 41.35 | 29.22 | 47.58 | 26.95 | 44.62 |
| | Ours+Slerp | ✓ | 32.24 | 49.12 | 23.03 | 45.56 | 33.76 | 54.61 | 29.68 | 49.76 |
| | Δ (Ours vs Slerp) | | (+4.66) | (+6.23) | (+1.61) | (+4.21) | (+4.54) | (+7.03) | (+2.73) | (+5.14) |
| | Slerp+TAT (ECCV'24) | | 29.64 | 46.47 | 23.35 | 45.12 | 31.97 | 51.20 | 28.32 | 47.60 |
| | LDRE (SIGIR'24) | ✓ | 31.04 | 51.22 | 22.93 | 46.76 | 31.57 | 53.64 | 28.51 | 50.54 |
| | CIReVL (ICLR'24) | ✓ | 29.49 | 47.40 | 24.79 | 44.76 | 31.36 | 53.65 | 28.55 | 48.57 |
| | ImageScope (WWW'25) | ✓ | 27.82 | 41.76 | 20.18 | 37.48 | 28.61 | 44.42 | 25.54 | 41.22 |
| | PrediCIR (CVPR'25) | | 31.80 | 52.00 | 25.40 | 49.50 | 33.10 | 55.40 | 30.10 | 52.30 |
| | OSrCIR (CVPR'25) | ✓ | 33.17 | 52.03 | 29.7 | 51.81 | 36.92 | 59.27 | 33.26 | 54.37 |
| | MMRet-Large (ACL'25)† | | 37.04 | 56.13 | 29.84 | 50.66 | 37.07 | 59.01 | 34.65 | 55.27 |
| | Ours+MMRet-large | ✓ | 39.1 | 58.34 | 31.33 | 52.35 | 39.67 | 61.65 | 36.70 | 57.45 |
| | Δ (Ours vs MMRet-Large) | | (+2.06) | (+2.21) | (+1.49) | (+1.69) | (+2.6) | (+2.64) | (+2.05) | (+2.18) |

## 4.2 QUALITATIVE ANALYSIS

To more intuitively demonstrate the effectiveness of our method, we provide some successful retrieval cases in Figure 3. These cases cover a variety of complex modification texts, including: (a) For **action modification**, it relaxes overly specific constraints ("slim, long legs") to retrieve the target by focusing on general attributes ("tan dog"); (b) For **attribute and number modification**, it integrates gradual refinements ("remove one dog, add grass, make one dog black") to incrementally improve retrieval accuracy; (c) For **quantity and scene change**, it resolves semantic conflicts (e.g., "savanna" vs. "hillside") by generalizing constraints ("savanna" to "landscape"); (d) For **holistic replacement**, it handles substantial semantic gaps ("blue pouch" to "yellow shoe") by isolating the query's core intent, enabling successful retrieval. More detail analysis can be found in Section A.5. These successful cases demonstrate that our iterative approach effectively handles complex modifi-

Figure 3: Examples from the CIRR validation set where our method retrieves the desired image, in comparison to the MMRet-large baseline. Our approach utilizes two additional rounds (Round 1, Round 2) of iterative synthetic caption generation to refine the retrieval process. The green box indicates the ground truth image. Within the synthetic captions, green highlighting marks correct key visual elements, while pink highlighting denotes elements irrelevant to the target image.

cations including attribute changes, scene replacements, quantity adjustments, and other combinatorial edits, significantly improving retrieval performance through self-correcting refinement.

## 4.3 ABLATION STUDY

Table 3: Ablation study on CIRCO and FashionIQ validation data using MMRet-Large as the baseline. "reflection" indicates that the MLLM reflects using retrieved images from the previous round; "query fusion" refers to the historical query fusion strategy; "random caption" means that the captions provided to the MLLM are randomly generated by captioner; "optimized caption" means that the captions are generated by our strategies.

| Method | CIRCO | | | | Fashion-IQ | |
|---|---|---|---|---|---|---|
| | k=5 | k=10 | k=25 | k=50 | k=10 | k=50 |
| *Impact of Different Components* | | | | | | |
| baseline | 37.75 | 38.60 | 41.22 | 42.09 | 34.65 | 55.27 |
| reflection | 36.66 | 37.09 | 39.69 | 40.59 | 30.82 | 50.71 |
| reflection + query fusion | 40.82 | 41.26 | 43.83 | 44.79 | 36.26 | 57.04 |
| reflection + query fusion + random caption | 41.85 | 42.35 | 44.76 | 45.69 | 36.35 | 57.10 |
| reflection + query fusion + optimized caption (full model) | 42.77 | 43.08 | 45.64 | 46.62 | 36.70 | 57.45 |
| *Impact of Different Prompt Strategies* | | | | | | |
| w/o CoT | 41.45 | 41.88 | 44.37 | 45.36 | 35.68 | 56.38 |
| w/o think process | 40.77 | 40.99 | 43.66 | 44.54 | 35.97 | 56.57 |
| *Impact of MLLM Choice* | | | | | | |
| Qwen-2.5VL-3B | 39.40 | 39.82 | 42.38 | 43.32 | 34.88 | 55.50 |
| Qwen-2.5VL-7B | 39.31 | 40.11 | 42.54 | 43.57 | 34.72 | 55.22 |
| Qwen-2.5VL-72B | 41.50 | 42.12 | 44.59 | 45.52 | 36.40 | 56.99 |

To dissect the contribution of each component and design choice within our framework, we conduct a series of ablation studies.

**Impact of Different Components and Prompt Strategies.** As shown in the Table 3, we metric the impact of different components on performance. It can be seen that the performance degradation without historical query fusion highlights its critical role in preventing "query drift" and maintaining retrieval stability. Furthermore, incorporating captions as contextual information enhances the retrieval performance, and the captions optimized through our strategy provide greater benefits compared to randomly selected captions. Additionally, we separately evaluated performance under conditions where the model is allowed to reason independently without using carefully prepared CoT templates("w/o CoT"), and where it is instructed to output the final answer directly without displaying its reasoning process("w/o think process"), to demonstrate the validity of our design.

**Impact of MLLM Choice.** To verify the generalizability of our framework, we evaluate its performance through different open source MLLMs. As shown in the Table 3, our method achieves

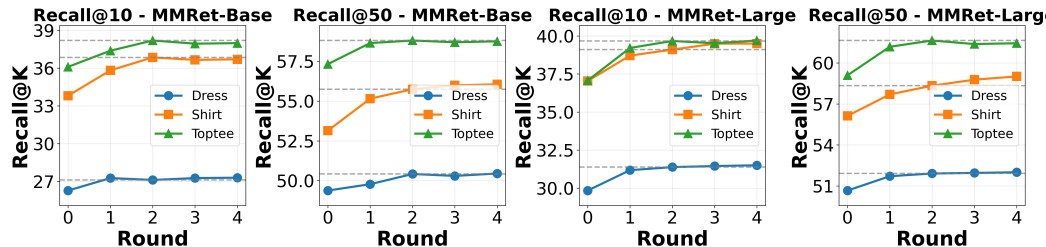

Figure 4: Recall@K Performance Analysis Across Rounds on FashionIQ Validation Data.

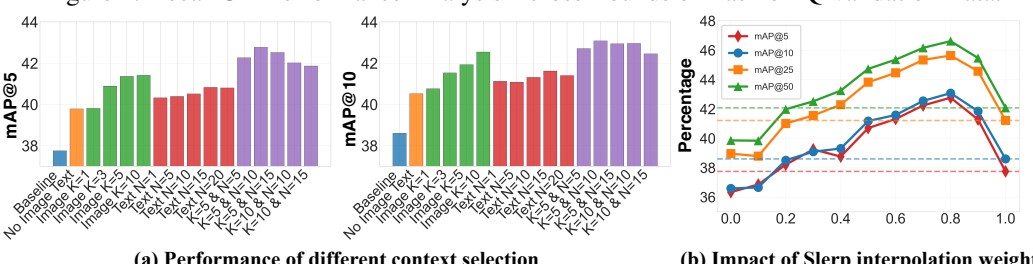

(a) Performance of different context selection     (b) Impact of Slerp interpolation weight

Figure 5: Performance comparison of different context selection strategies and Impact of Slerp interpolation weight on CIRCO validation data.

significant performance gains even with relatively small models like Qwen-2.5VL-7B. Moreover, the performance improvement becomes more pronounced as the model size increases.

**Impact of different stop round.** In Section 4.1 we report the results of adding an additional two rounds. Here we give the results for more rounds in the Figure 4. As can be seen, the gains become very low when going beyond 2 rounds, and for cost reasons we consider 2 rounds to be the best stopping point. For the complete ablation study, please refer to the Appendix in Section A.4. Additionally, since our framework is a multi-round iterative process, we also investigate the temporal cost of our method and more diverse stopping conditions. Each additional iteration adds approximately 3 seconds of latency, yet the overall runtime remains competitive compared to other methods. Please refer to Section A.7 and Section A.8 in the supplementary materials.

**Impact of Top-K Images and Top-N Captions.** As shown in Figure 5 (a), we investigated the effect of varying the number of retrieved top-K images and top-N captions as the context for MLLM. The results reveal that K=5 and N=10 achieves the highest scores across all metrics. We can find that using too few images and captions leads to a lack of information for effective reasoning, while using too many introduces noise from irrelevant features that can degrade retrieval accuracy. For the complete ablation study, please refer to the Appendix in Section A.4.

**Impact of Different Slerp Iterpolation Weight.** As shown in Figure 5 (b), we analyze the impact of the interpolation weight $\alpha$ (as defiend in Equation 3) on model performance. The horizontal axis represents the value of $\alpha$. As $\alpha$ increases, the proportion of the global vector increases. It can be seen that the best performance is achieved when $\alpha = 0.8$.

## 5 CONCLUSION

In this paper, we introduce CoRR, a novel training-free framework that addresses the fundamental limitation of existing ZS-CIR methods. Specifically, CoRR introduces a closed-loop "retrieval-reflection-refinement" framework. It employ an MLLM to analyze issues in the previously retrieved results by Chain-of-Thought reasoning and refine queries iteratively for improved accuracy, while mechanisms like Slerp-based query fusion and retrieval-driven contextual optimization ensure stable refinement and alignment with the retrieval model. By achieving SOTA performance across multiple benchmarks, CoRR highlights the potential of iterative retrieval strategies guided by self-reflection. In the future, we may extend CoRR to other domains, such as video retrieval or multi-step reasoning tasks, and further enhance its adaptability in complex scenarios.

## REPRODUCIBILITY STATEMENT

We demonstrate the iterative procedure and module designs in Section 3. The complete Chain of Thought prompt template used by the MLLM is provided in Appendix Section A.1. Datasets and evaluation protocols are specified in the Section 4, and the main quantitative results are summarized in Section 4.1. Comprehensive ablations and sensitivity analyses appear in Section 4.3 and Appendix Section A.4, with additional results in Appendix Section A.2 and qualitative analyses in Section 4.2 and Appendix Section A.3. These referenced sections collectively specify the method, experimental settings, and evaluation details necessary to reproduce the reported findings.

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

# A    APPENDIX

## A.1    COMPLETE TEMPLATE FOR COT PROMPT

You are an expert in visual understanding and iterative image retrieval.
Your task is to analyze the current retrieval results and generate a better target image caption to improve retrieval accuracy in the next round.
You should think step by step:
**# Guidelines**
**## Step 1. Understand User Intent**
  - Begin by analyzing the original image to capture all visible objects, attributes, and elements, including specific details such as object types, relationships, colors, scenes, and the overarching domain of the image.
  - Carefully interpret the user's intent described in the Original Query, considering semantic aspects such as addition, negation, spatial relations, or viewpoint changes. Explain how these intents are translated into the target image description.

**## Step 2. Analyze Retrieval Results**
  - Briefly examine the visual patterns present in the retrieved images and captions, focusing on recurring objects, attributes, quantities, relationships, and other defining features. Identify these elements as they reflect the retrieval model's preferences and help filter out irrelevant noise from the images.
  - Compare the recurring elements in the retrieved results against the user's intent. Identify key visual elements that are consistently missing, misrepresented, or incorrect in the captions, as well as those that correctly align with the intended modification, such as specific items, properties, counts, poses, or spatial relationships.
  - Highlight commonalities across the retrieved results to pinpoint trends or gaps in alignment with the intended modification. Focus on how these shared characteristics deviate from or meet the user's expectations.

**## Step 3. Problem Reflection**
  - Influence of Manipulation Intent: Clearly identify how the modification request impacted your approach to adapting the original image description. Focus on specific instructions, such as adding, removing, or altering certain elements, and the corresponding semantic transformations.
  - Analysis of Current Errors: Based on the visual and textual aspects of the retrieved results, explain why these results fail to fully implement the requested modification. Highlight critical visual elements that were missing, ambiguous, or inaccurately described in the current top images or captions.
  - Semantic and Visual Adjustments: Clarify what specific visual features, relationships, or spatial details need to be emphasized to achieve alignment with the requested transformation. If ambiguities exist in the query or description, explain how they influenced these retrieval errors and how they can be resolved.

**# Generation Strategy**
**## Caption Style Alignment**
  - Study the top retrieved candidate image captions to understand the common style features. Provide specific observations in the following structured format:
  - The typical length and structure: How long are the captions? Are they short and direct or longer and more descriptive?
  - The level of detail: What level of specificity is used? Do captions focus on high-level actions (e.g., "pulling") or include fine-grained details ?
  - Common vocabulary and phrasing patterns: How are objects, colors, and spatial relationships described?
  - Ensure the new caption aligns with these observed style features.

**## Caption Generation Strategy**
  - Provide a target image caption that captures the user's intent and the key visual constraints.
  - De-emphasize already satisfied aspects unless discriminative.
  - Prefer generalizable, descriptive terms; avoid dataset-specific IDs or rare jargon.
  - Based on the Candidate Analysis and identified failure modes, decide which terms to keep, drop, relax, or rephrase.
  - **Avoid self-referential language**: Do not generate captions with references like "this image" or "it." Focus on standalone descriptive statements.
  - **Avoid negation**: Replace negative expressions with affirmative descriptions. Instead of saying "without clutter," describe the intended cleanliness or simplicity (e.g., "neatly arranged").

**# Output Format**
  - Keep it concise and factual, avoiding overly creative or detailed descriptions
  - First, please provide the most concise think steps within the <think> </think> tags. Ensure each step is brief, direct, and free from unnecessary details or redundant information.
  - Then, provide your optimized target caption in <answer> </answer> tags

Figure 6: Complete prompt template.

Figure 6 presents our reflective CoT prompt template for iterative retrieval. This template guides the MLLM through a systematic four-step reasoning process:

**Step 1: Understand User Intent.** The template first analyzes the original image to capture all visible objects, attributes, and spatial relationships, then carefully interprets the user's modification

intent (addition, negation, spatial changes, etc.) and explains how these translate into target image requirements.

**Step 2: Analyze Retrieval Results.** The template examines visual patterns in the Top-K retrieved images and Top-N captions, identifying recurring elements that reflect the retrieval model's preferences. It compares these patterns against user intent to pinpoint missing, misrepresented, or correctly aligned visual elements, highlighting commonalities and gaps in modification alignment.

**Step 3: Problem Reflection.** The template performs three key analyses: (i) *Influence of Manipulation Intent*: identifying how modification requests impact the approach; (ii) *Analysis of Current Errors*: explaining why retrieved results fail to implement the requested modification; and (iii) *Semantic and Visual Adjustments*: clarifying specific visual features that need emphasis to achieve proper alignment.

**Step 4: Caption Generation Strategy.** The template guides the MLLM to study retrieved captions and learn their style characteristics, including typical length, detail level, and vocabulary patterns. It then instructs the MLLM to generate a target caption that preserves user intent while incorporating key visual constraints, uses generalizable descriptive terms, avoids self-referential language and negation, and aligns with the learned linguistic style to maximize retrieval effectiveness.

Finally, The template constrain the model outputs structured reasoning in `<think>` tags followed by the optimized caption in `<answer>` tags.

## A.2 MORE RESULTS

Due to space constraints, we present additional results on the GeneCIS dataset in Table 4. Our method achieves state-of-the-art performance with an average R@1 score of 19.53, demonstrating the effectiveness of our iterative retrieval approach. Notably, we outperform SOTA methods such as OSrCIR by 1.63 points, highlighting the significant improvement brought by our iterative reflective CoT mechanism and Retrieval-Driven Caption Optimization strategy.

Table 4: **Evaluation on GeneCIS Test Data.** Methods with "†" are our implementations. Green highlighting indicates the best performance, while blue highlighting indicates the second-best performance.

| | Architecture | Focus Attribute | | | Change Attribute | | | Focus Object | | | Change Object | | | Avg. |
|---|---|---|---|---|---|---|---|---|---|---|---|---|---|---|
| | | R@1 | R@2 | R@3 | R@1 | R@2 | R@3 | R@1 | R@2 | R@3 | R@1 | R@2 | R@3 | R@1 |
| CLIP-ViT-L | SEARLE-XL (ICCV'23) | 17.00 | 29.70 | 40.70 | 16.40 | 25.30 | 34.10 | 8.00 | 16.90 | 25.60 | 7.90 | 16.80 | 24.80 | 12.30 |
| | LinCIR (CVPR'24) | 16.90 | 30.00 | 41.50 | 16.20 | 28.00 | 36.80 | 8.30 | 17.40 | 26.20 | 7.40 | 15.70 | 25.00 | 12.20 |
| | CIReVL (ICLR'24) | 19.50 | 31.80 | 42.00 | 14.40 | 26.00 | 35.20 | 12.30 | 21.80 | 30.50 | 17.20 | 28.90 | 37.60 | 15.90 |
| | OSrCIR (CVPR'25) | 20.90 | 33.10 | 44.50 | 17.20 | 28.50 | 37.90 | 15.00 | 23.60 | 34.20 | 18.40 | 30.60 | 38.30 | 17.90 |
| | MMRet-Large (ACL'25)† | 18.95 | 29.94 | 39.55 | 14.59 | 26.75 | 36.41 | 16.87 | 25.82 | 36.53 | 18.12 | 31.08 | 39.18 | 17.13 |
| | **Ours+MMRet-Large** | **21.65** | 31.65 | 41.55 | **17.57** | **28.60** | 37.17 | **19.10** | 27.41 | 37.82 | **19.81** | **32.31** | **41.43** | **19.53** |
| | Δ *(Ours vs MMRet-Large)* | *(+2.70)* | *(+1.71)* | *(+2.00)* | *(+2.98)* | *(+1.85)* | *(+0.76)* | *(+2.23)* | *(+1.59)* | *(+1.29)* | *(+1.69)* | *(+1.23)* | *(+2.25)* | *(+2.40)* |

Additionally, we present results on more diverse model configurations in Table 5 to demonstrate the generalization capability and universality of our method. To ensure fairness and consistency across evaluations while minimizing computational costs, we use the CLIP-ViT-L/14 architecture as the backbone, with weights initialized from OpenAI's pretrained model (Radford et al., 2021). we adopt Qwen-VL-Max as the reasoning model throughout the experiments. For CIReVL, we adopt BLIP-2 (Li et al., 2023a) with a OPT-6.7b (Zhang et al., 2022) language model as the image captioner. Our approach consistently improves performance across different backbone models, validating its plug-and-play nature and broad applicability.

## A.3 QUALITATIVE ANALYSIS OF RETRIEVAL-DRIVEN CAPTION OPTIMIZATION STRATEGY

Figure 7 demonstrates the necessity of our Retrieval-Driven Caption Optimization strategy through qualitative analysis. The figure reveals that different generated captions yield significantly different retrieval results, indicating that embedding models exhibit strong preferences for specific linguistic formulations and visual element descriptions.

For instance, in the first example, the retrieval model requires explicit mention of "chips" rather than the generic term "snacks" to achieve accurate retrieval. This demonstrates that embedding models are sensitive to the level of specificity and vocabulary choice in captions.

Table 5: **More Results on FashionIQ Test Data.** Results are grouped by architecture and sorted by publication year. Training-free methods are marked with "✓". Methods with "†" are our implementations. Our method consistently improves over baseline methods across different architectures.

| | Architecture | Training-free | Shirt | | Dress | | Toptee | | Average | |
|---|---|---|---|---|---|---|---|---|---|---|
| | | | R@10 | R@50 | R@10 | R@50 | R@10 | R@50 | R@10 | R@50 |
| CLIP-ViT-L | LinCIR (CVPR'24)† | | 28.26 | 46.57 | 20.58 | 41.99 | 28.56 | 49.36 | 25.80 | 45.97 |
| | Ours+LinCIR | ✓ | 32.14 | 50.64 | 22.31 | 43.33 | 32.23 | 52.78 | 28.89 | 48.92 |
| | Δ (Ours vs LinCIR) | | (+3.88) | (+4.07) | (+1.73) | (+1.34) | (+3.67) | (+3.42) | (+3.09) | (+2.95) |
| | CIReVL (ICLR'24)† | ✓ | 15.01 | 24.48 | 10.70 | 25.80 | 13.10 | 26.05 | 12.94 | 25.44 |
| | Ours+CIReVL | ✓ | 19.97 | 32.67 | 14.37 | 31.73 | 19.63 | 36.10 | 17.99 | 33.50 |
| | Δ (Ours vs CIReVL) | | (+4.96) | (+8.19) | (+3.67) | (+5.93) | (+6.53) | (+10.05) | (+5.53) | (+8.06) |
| | OSrCIR (CVPR'25)† | ✓ | 26.84 | 43.77 | 16.46 | 34.95 | 24.63 | 43.65 | 22.64 | 40.79 |
| | Ours+OSrCIR | ✓ | 29.49 | 47.15 | 19.63 | 39.81 | 28.96 | 49.21 | 26.03 | 45.39 |
| | Δ (Ours vs OSrCIR) | | (+2.65) | (+3.38) | (+3.17) | (+4.86) | (+4.33) | (+5.56) | (+3.38) | (+4.60) |

Our optimized captions not only provide well-aligned examples for the MLLM but also help it identify the most discriminative visual elements in retrieved images, enabling more effective iterative refinement and ultimately improving retrieval accuracy.

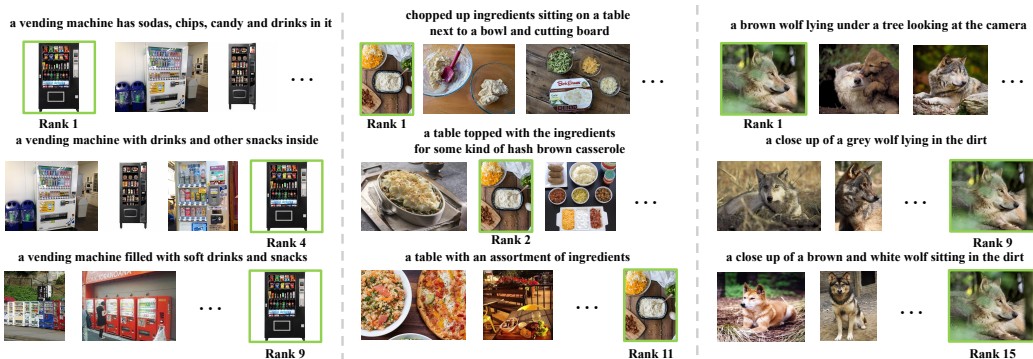

Figure 7: Visual Samples of Retrieval-Driven Caption Optimization.

## A.4 MORE ABLATION STUDIES

Table 6: Ablation study on FashionIQ validation dataset across different categories. "reflection" indicates that the MLLM reflects using retrieved images from the previous round; "query fusion" refers to the historical query fusion strategy; "random caption" means that the captions provided to the MLLM are randomly generated by captioner; "optimized caption" means that the captions are generated by our strategies.

| Method | Dress | | Shirt | | TopTee | |
|---|---|---|---|---|---|---|
| | R@10 | R@50 | R@10 | R@50 | R@10 | R@50 |
| *Impact of Different Components* | | | | | | |
| baseline | 29.84 | 50.66 | 37.04 | 56.13 | 37.07 | 59.01 |
| reflection | 25.12 | 46.49 | 33.72 | 51.21 | 33.62 | 54.42 |
| reflection + query fusion | 31.28 | 51.76 | 38.79 | 58.22 | 38.72 | 61.14 |
| reflection + query fusion + random caption | 31.17 | 52.04 | 38.74 | 58.28 | 39.13 | 60.97 |
| reflection + query fusion + optimized caption (full model) | 31.33 | 52.35 | 39.10 | 58.34 | 39.67 | 61.65 |
| *Impact of Different Prompt Strategies* | | | | | | |
| w/o CoT | 30.15 | 51.78 | 38.53 | 57.41 | 38.36 | 59.94 |
| w/o think process | 30.68 | 51.51 | 38.45 | 57.39 | 38.77 | 60.81 |
| *Impact of Different Components* | | | | | | |
| Qwen-2.5VL-3B | 29.74 | 49.82 | 37.48 | 56.72 | 37.43 | 59.96 |
| Qwen-2.5VL-7B | 29.79 | 49.62 | 36.27 | 56.37 | 38.09 | 59.66 |
| Qwen-2.5VL-72B | 31.03 | 52.05 | 39.15 | 58.19 | 39.01 | 60.73 |

Table 6 presents detailed ablation study results on the FashionIQ dataset across three categories (Dress, Shirt, TopTee). The results demonstrate the effectiveness of each component in our framework, with our full method achieving the best performance across all categories and metrics.

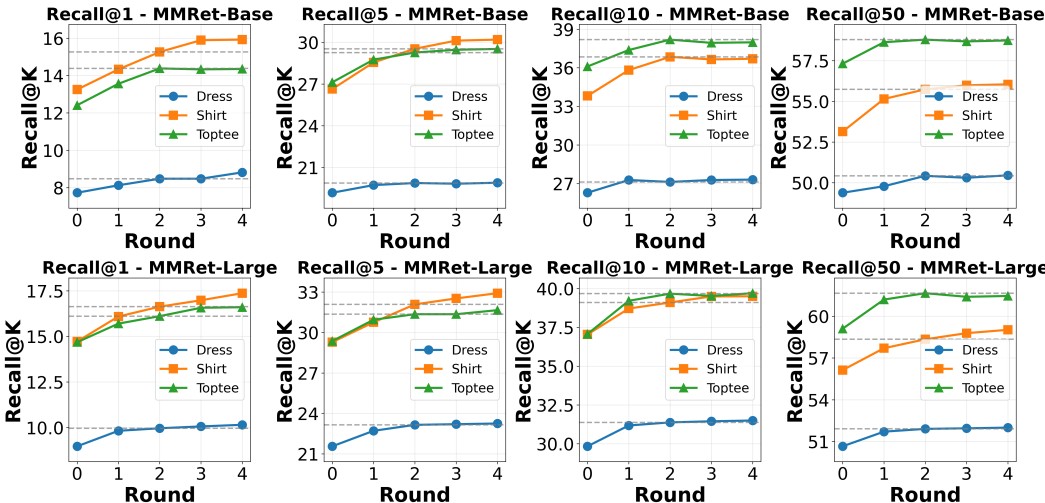

Figure 8: Recall@K Performance Analysis Across Rounds on FashionIQ validation data.

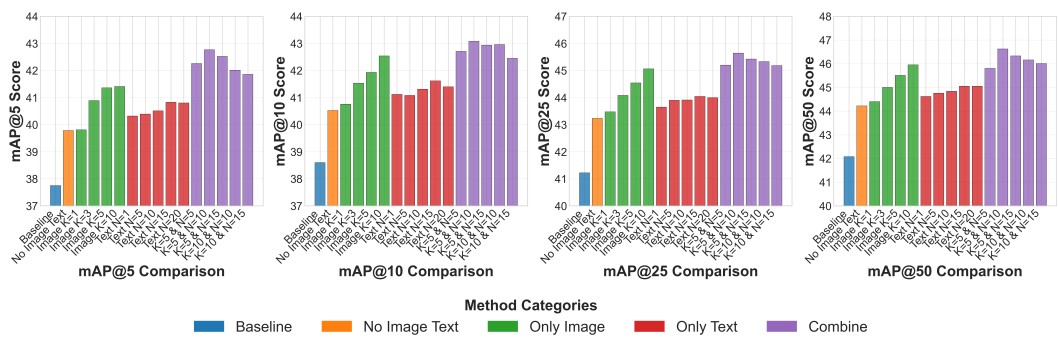

Figure 9: Performance comparison of different context selection on CIRCO validation data.

In Section 4.3, we briefly analyzed the impact of different rounds on performance. Here, we present the complete ablation study results in Figure 8. It can be seen that Recall@1 and Recall@5 still show significant improvement after two additional rounds.

In Section 4.3, we briefly analyzed the impact of different context selection strategies for retrieved results on performance. Here, we present the complete experimental results in Figure 9.

Additionally, we employ slerp as the strategy for historical query fusion. We present the results under linear interpolation to demonstrate slerp's effectiveness. In addition, we also update the query representation using the mean embedding of the top-5 retrieved images to simulate a pseudo-relevance feedback strategy. As shown in Figure 7, linear interpolation only yields a marginal performance improvement, whereas a simple pseudo-relevance feedback strategy degrades performance due to the noise it introduces.

Table 7: Ablation study on FashionIQ validation dataset across different query fusion strategies.

| Method | Dress | | Shirt | | TopTee | |
|---|---|---|---|---|---|---|
| | R@10 | R@50 | R@10 | R@50 | R@10 | R@50 |
| baseline | 29.84 | 50.66 | 37.04 | 56.13 | 37.07 | 59.01 |
| linear interpolation | 28.84 | 50.72 | 38.17 | 57.70 | 37.97 | 60.08 |
| pseudo-relevance feedback | 27.56 | 46.05 | 32.82 | 48.77 | 33.45 | 52.12 |
| proposed | 31.33 | 52.35 | 39.10 | 58.34 | 39.67 | 61.65 |

## A.5 ANALYSIS OF SUCCESSFUL CASES

In Section 4.2, we provide some successful retrieval cases and simple analysis of these cases. Here we provide more detailed analysis of these cases.

**Action Modification (top-left)**: In Round 1, our model correctly incorporates "sits on the wood floor" but retains source image attributes "slim" and "long legs." This over-specific query fails retrieval as no candidate images match this exact combination. In Round 2, the model recognizes these attributes are too restrictive and strategically drops them, focusing on the more general "tan" attribute (also from the source image). The refined query "a tan dog sitting on a wooden floor" successfully retrieves the target, demonstrating our method's ability to self-correct by relaxing overly specific constraints.

**Quantity and Scene Change (bottom-left)**: The static baseline fails by over-emphasizing the "savanna" element from the reference image while ignoring the "antelopes" concept. In Round 1, our model correctly identifies "a herd of antelopes" and "thorny" texture, but incorrectly retains "savanna" as the environment. The ground truth shows antelopes on a hillside, not flat savanna plains, causing retrieval failure due to semantic over-specification. In Round 2, the model identifies "savanna" as the conflict point and strategically generalizes it to "landscape." This relaxation preserves essential elements while resolving the inconsistency, successfully retrieving the correct image. This demonstrates our method's ability to self-correct by relaxing overly constrained descriptors.

**Attribute and Number Modification(top-right)**: The baseline completely fails on this complex query requiring quantity change ("remove one dog"), background modification ("add grass"), and attribute alteration ("make one dog black"). In Round 1, our model correctly identifies the specific breed ("Cavalier King Charles Spaniels") and successfully retrieves the target among top-5 candidates. In Round 2, the model further refines the description by adding "brown and white" coloring for the second dog, promoting the target to the top position. This demonstrates our method's ability to achieve initial success and then enhance precision through iterative refinement.

**Holistic Replacement(bottom-right)**: Unlike the previous examples, the baseline fails due to semantic irrelevance between source image (blue pencil case) and modification text ("yellow coloured shoe"), resulting in irrelevant retrievals. In Round 1, our model isolates the core intent and generates "a yellow shoe placed on a white background," disregarding the unrelated source context. In Round 2, the model refines to "a bright yellow running shoe with black laces and white sole," successfully retrieving the target. This demonstrates our method's robustness against substantial semantic gaps that severely hinder baseline performance.

## A.6 ANALYSIS OF FAILURE CASES

Since our approach is entirely training-free, it still requires improvement in semantic accuracy, contextual sensitivity, and visual-language alignment under certain scenarios. We focus on analyzing cases where performance actually declined after query enhancement by our method. As shown in Figure 10, these are partial failure cases from our testing on the CIRR validation dataset using MMRet-Large.

First, the case in the upper left corner demonstrates that when MLLM is tasked with analyzing initial retrieval results and optimizing queries based on them, the model may introduce noise attributes unrelated to the original image (such as erroneously adding "white" to the textual description), causing subsequent retrievals to deviate from the intended target. Secondly, the case in the lower left corner demonstrates that when the modification instructions deviate significantly from the original image, MLLM still relies on the original image content when generating enhanced queries. This failure to adequately reflect user intent leads to semantic confusion. Third, the case in the upper right corner reveals the model's limitations in understanding fine-grained semantics, such as misinterpreting "lighter" as "lighter-colored dog", reflecting a bias in understanding abstract visual concepts. Finally, in the case study at the bottom right, the model erroneously retained a prominent element (a baby pelican) that should have been modified or ignored in the original image, indicating it still faces challenges in performing selective editing and attention control.

## A.7 EFFECTIVENESS AND EFFICIENCY ANALYSIS

As shown in the Figure 8, the average processing time for our additional two iterations is 6.44 seconds, with API calls accounting for 98% of this duration. This is slightly slower than CIReVL (2.21 seconds) but remains substantially faster than the other methods. When we remove the reasoning process and only output the final answer ("w/o think process") as shown in Table 3, the average

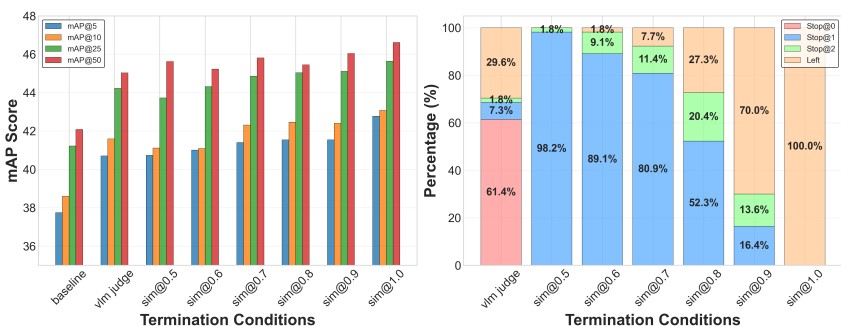

Figure 10: Visualization of failure cases. The first row shows the original image, modification text, and the baseline retrieval results; the second row shows our synthesized captions and the corresponding retrieval results.

Figure 11: Different termination conditions.

processing time reduces to 3.58 seconds at the cost of a slight performance degradation. We believe more efficient APIs in the future can resolve this trade-off between processing speed and reasoning quality.

Table 8: Average processing time per query. For a fair comparison, we use Qwen-VL-Max as the reasoning model and BLIP-2 as the captioner, on a single NVIDIA A6000 48GB GPU.

| Method | CIReVL (ICLR' 24) | LDRE (SIGIR' 24) | OSrCIR (CVPR' 25) | Proposed |
|---|---|---|---|---|
| Seconds | 2.21 | 14.26 | 13.38 | 6.44 |

## A.8 DIFFERENT TERMINATION CONDITIONS

In the main text, we present the results of conducting two additional rounds of reflection. In fact, we also explore more diverse stopping conditions, but none yielded better results than simply running the two rounds to completion.

As shown in Figure 11, left is the performance comparison of different methods on CIRCO validation data, where sim@k means the iteration stops when the similarity between the new query and global query exceeds k. vlm judge means the Qwen-VL-Max gives a judgment on whether to continue the iteration. The right shows the proportion of samples that cease to proceed to the next round after a certain number of rounds under different methods.

As can be seen, the best results are achieved when all samples complete two additional rounds, though this approach incurs higher costs. Using VLM as the judge yields the poorest performance. sim@0.7 ensures that 80% of samples terminate after Round 1 while maintaining relatively good performance.

## A.9 LLM Clarification

We clarify the role of Large Language Models (LLMs) in the preparation of this manuscript. Specifically, LLMs were used to refine language quality by correcting grammatical errors, improving sentence structure, and enhancing the overall clarity, coherence, and flow of the text.

