# OpenReview forum: "Reflection from Retrieval: MLLM-Guided Iterative Reasoning for Zero-Shot Composed Image Retrieval"
_ICLR.cc/2026/Conference — ICLR 2026 Conference Withdrawn Submission_

### Official Review · Reviewer_K6DJ · 2025-10-28

**Soundness:** 3
**Presentation:** 3
**Contribution:** 2
**Rating:** 4
**Confidence:** 4

**Summary:**

This paper proposes CoRR, a training-free framework to address ZS-CIR task that enables dynamic, self-correcting query refinement. It first utilizes MLLM for reasoning and self-reflection. Then, this paper designs a Slerp-based historical query fusion and retrieval-driven caption optimization strategy for progressive alignment with user intent and retrieval models. Experiments on multiple ZS-CIR benchmarks demonstrate the superior performance of CoRR.

**Strengths:**

1. The proposed CoRR method improves several current ZS-CIR baselines, and achieves superior performance compared with previous SOTA methods.
2. The ablation study is sufficient to support the stated observations.

**Weaknesses:**

1. Insufficient Technical Contribution. This paper provides limited technical contribution that most of the techniques follow previous works.
1) The reasoning and self-reflection process attempts to mine the reasoning capability of MLLMs, which is very similar to works like OSrCIR [1].
2) The historical query fusion utilizes SLERP as the integration technique between embeddings, which is similar with Slerp [2].
3) The Retrieval-Driven Caption Optimization simply generates captions via BLIP-2 with a selection strategy, which is similar with SEIZE [3].

[1] Tang, Yuanmin, et al. "Reason-before-retrieve: One-stage reflective chain-of-thoughts for training-free zero-shot composed image retrieval." Proceedings of the Computer Vision and Pattern Recognition Conference. 2025.
[2] Jang, Young Kyun, et al. "Spherical linear interpolation and text-anchoring for zero-shot composed image retrieval." European Conference on Computer Vision. Cham: Springer Nature Switzerland, 2024.
[3] Yang, Zhenyu, et al. "Semantic editing increment benefits zero-shot composed image retrieval." Proceedings of the 32nd ACM International Conference on Multimedia. 2024.

Without emphasizing novel motivations, this paper seems to stack previous works to boost the performance of ZS-CIR. The underlying reason mostly originates from the multiple retrieved images with synthetic captions.

2. Time cost. This paper utilizes a strong MLLM to to generate the thinking results and the target caption. Also, a BLIP-2 captioner is utilized to generate M captions with further selection process for each of the Top-K retrieved images. Obviously, it costs heavy computational costs compared with previous methods, especially when N and K are large. Therefore, a quantitative comparison of inference costs between these related previous methods should be presented.

3. Stated as a plug-and-play method, can the proposed method be implemented upon baselines like OSrCIR, which already employs a MLLM reasoning process? Will the performance be better? What about on the textual inversion methods, such as Pic2Word?

4. Difference with previous methods. As many previous works, such as OSrCIR, have focused on the reasoning capability of MLLM, how the proposed CoT prompt differs from those proposed in previous works? Will the performance be better with their CoT prompt design?

5. Why can the proposed historical query fusion based on slerp better integrate different embeddings? How the Slerp mechanism reweights between historical and current embeddings? It’ll be better to provide a visualization on how the weight value of slerp during different iterations, and provide more in-depth analysis on why slerp plays an important part. What about using the simple EMA technique?

6. Concerns on generalizability of MLLM. It seems when using Qwen-2.5VL-7B and 3B, the performance degrades to 39.31, 39.40 compared with the baseline 40.20 (MMRet-Large) as stated in Table 1. This phenomenon shows that the proposed method has very poor robustness on different choice of MLLM.

**Questions:**

Please refer to weaknesses.

---

> ### Author Response · Authors · 2025-11-19
> **Rebuttal for K6DJ**
>
> # Q1: Time cost
> The captions we synthesise are generated for images within the candidate pool rather than the query image. It is enabling offline preprocessing. This is feasible because in practical scenarios, the retrieval pool typically remains constant. During actual inference, a single query will only invoke the MLLM for self-reflection. We present a performance comparison of different methods in Appendix A.7.
>
> # Q2: he proposed method be implemented upon baselines like OSrCIR
> The relevant results are presented in Appendix A.2. These include LinCIR, CIReVL, and OsrCIR.
>
> # Q3: how the proposed CoT prompt differs from those proposed in previous works？
> Our most significant distinction lies in the fact that our CoT reflects upon actual retrieval results, whereas previous methods merely analysed the query. Thanks to retrieval feedback, MLLM can access more relevant information for analysis and perceive aspects of the retrieval pool, rather than relying solely on query-based inference.
>
> # Q4: Why can the proposed historical query fusion based on slerp better integrate different embeddings?
> We have presented the ablation results for linear interpolation and pseudo-relevance feedback in Appendix A.4.
>
> # Q5: Concerns on generalizability of MLLM.
> Note that Table 1 presents the test data, while Table 3 presents the validation data

---

### Official Review · Reviewer_3hxB · 2025-10-30

**Soundness:** 2
**Presentation:** 3
**Contribution:** 2
**Rating:** 4
**Confidence:** 4

**Summary:**

In this paper, the authors proposed a training-free composed image retrieval framework. The framework is based on MLLMs and basic retrieval models. Specifically, the authors propose to optimize the composed caption by analyzing retrieved results, problem reflection, and a novel generation strategy. In addition, they propose a retrieval-driven caption optimization strategy to further improve the retrieval quality. Experiments are conducted on CIRR, CIRCO, FashionIQ and GeneCIS datasets.

**Strengths:**

1. The paper presentation looks good and the methodology part seems reasonable.

2. Experimental results look promising and multiple ablations are conducted.

**Weaknesses:**

1. The efficiency is significantly concerning. In each retrieval process, there are two MLLMs involved. Each MLLM is used for multiple rounds of generation, which may make the entire generation process extremely slow. In fact, I am concerned whether the proposed method is practical in real-world applications because both composed and target sides require MLLMs and the multiple generation may significantly undermine many time-sensitive retrieval applications.

2. The optimization process can hardly be guaranteed. The self-reflection process updates the composed caption. Yet, it is unclear whether the update is beneficial. In addition, whether the  composed caption can directly benefit the retrieval is unclear as well. The method itself lacks guarantee that the optimization will be conducted towards minimizing the loss.

3. Experiments only tested the proposed method together with Slerp or MMRet. However, its combination with the original CLIP is not discussed in the main paper. It is recommended to include the comparison using the original CLIP (and include the ViT-G version) to compare with other baselines.

**Questions:**

See the weakness above

---

> ### Author Response · Authors · 2025-11-19
> **Rebuttal for 3hxB**
>
> # Q1: The efficiency is significantly concerning.
> During the inference process, we only invoke one MLLM and one embedding model. Note that the caption generation process can be pre-processed offline. Furthermore, in our experiments, we typically perform two additional iterations (primarily incurring the overhead of two extra MLLM API calls). This overhead is not significant; we provide a comparison of computational costs between our method and others in Appendix A.7.
>
> # Q2: The optimization process can hardly be guaranteed
> It is difficult to theoretically prove its effectiveness, as this relies heavily on the capabilities of the MLLM. From the results, it is evident that our approach enhances performance, but whether it represents the optimal method remains unverifiable. Nevertheless, it certainly warrants consideration.
>
>
> # Q3: Using the original CLIP to compare with other baselines
> Our approach functions more as a plug-and-play module that can be integrated into existing ZS-CIR methods; direct use of the original CLIP cannot achieve ZS-CIR. The simplest way to modify CLIP is based on Slerp, hence we present its performance metrics under Slerp. Furthermore, as Slerp-TAT did not report performance for ViT-G, we have omitted these figures as well. We will provide performance metrics in camera-ready.

---

### Official Review · Reviewer_ZGRB · 2025-10-30

**Soundness:** 2
**Presentation:** 3
**Contribution:** 2
**Rating:** 2
**Confidence:** 4

**Summary:**

This paper proposes CoRR, a training-free, reflective CoT-based framework for ZS-CIR. The proposed method leverage MLLM reasons over the top-K retrieved images conbine with BLIP-2–generated captions to produce multiple refined captions and then the new query is then updated via a Historical Query Fusion module that leverages Slerp to incorporate retrieval history from testing set. Integrated with CLIP-B/L, CoRR achieves SoTA results on CIRCO, CIRR, and FashionIQ.

**Strengths:**

1.	The idea of this paper is easy to understand.
2.	It is interesting to build a training-free iterative CIR model to refine intent understanding.

**Weaknesses:**

1.	Lack of novelty. Reflection on intermediate retrievals has been explored in prior works [1,2,3]. For training-free ZS-CIR specifically, ImageScope [1] already performs predicate verification and pairwise checks before refinement, while the paper does not compare in sufficient detail. Moreover, the Historical Query Fusion module appears to overlap with Slerp-TAT [4] (the described function and formulation seem nearly identical, i.e., Eq.(3) in the paper and Eq.(4) in Slerp-TAT [4]), but the paper does not clearly highlight the differences. This raises my concern about this paper's incremental novelty and missing attribution.
2.	Limit technology contribution. This paper's “MLLM-guided self-reflection” follows OSrCIR’s reflective CoT [5], but the authors do not acknowledge this explicitly and contrast the designs in their paper. Using BLIP-2 to generate multiple captions (in the Retrieval-Driven Caption Optimization module) builds on LDRE [6]. Moreover, using Slerp for composed retrieval follows Slerp-TAT [4]. Without a careful, side-by-side comparison and analysis, it is hard to make me to see sufficient new insights for ICLR.
3.	Potential test-time leakage. Historical Query Fusion combines the new query with history derived from test-time retrievals via Slerp. This mechanism risks implicitly encoding the testset distribution into the query. The choice of α=0.8 suggests the historical query plays a more important role than the new query, and Table 3 seems to show that history is the most influential component.
4.	Heavy pipeline and questionable efficiency comparisons. The method requires multiple MLLM calls and BLIP-2 caption sampling, substantially increasing inference cost. However, Table 8 reports only slight degradation while outperforming LDRE (BLIP-2 without reasoning) and OSrCIR (one-stage reasoning without BLIP-2). This comparison seems inconsistent with the added latency/complexity.
5.	Insufficient implementary details. The overall pipeline is complex, and the code are not given. For example, there are not sufficient details of how the Retrieval-Driven Caption Optimization module performs in the entire pipeline, and the details of calculating the retrieval-based validation approach are not enough. It is recommended to provide Pseudo-code.
6.	Concerns about the hallucination issues. The performance bis ased on  MLLM-generated results, which makes me concerned about the hallucination problem in the output. . A more detailed analysis of hallucination risks is needed.
7.	Insufficient ablation studies. For example, what is the performance of different query fusion methods rather than Slerp? What is the influence of different modules of their CoT method? What is the performance of different kind of MLLM (i.e., GPT, LLaVA)?
8.	Incomplete benchmarking. Comparisons on CLIP-G are missing, though it is common in recent ZS-CIR evaluations and the SoTA results for training-free ZS-CIR methods based on this backbone.

Overall, the novelty and technology contribution appear limited, and the Slerp-based method leverages test-set retrieval history, which may introduce data leakage, with key implementation details and hallucination analysis missing. Therefore, I gave the Reject recommendation, I believe the paper should have a revision to address these concerns.

References

[1] Luo P, Zhou J, Xu T, et al. ImageScope: Unifying Language-Guided Image Retrieval via Large Multimodal Model Collective Reasoning[C]//Proceedings of the ACM on Web Conference 2025. 2025: 1666-1682.

[2] Zhu H, Huang J H, Rudinac S, et al. Enhancing interactive image retrieval with query rewriting using large language models and vision language models[C]//Proceedings of the 2024 International Conference on Multimedia Retrieval. 2024: 978-987.

[3] Nara R, Lin Y C, Nozawa Y, et al. Revisiting relevance feedback for clip-based interactive image retrieval[C]//European Conference on Computer Vision. Cham: Springer Nature Switzerland, 2024: 1-16.

[4] Jang Y K, Huynh D, Shah A, et al. Spherical linear interpolation and text-anchoring for zero-shot composed image retrieval[C]//European Conference on Computer Vision. Cham: Springer Nature Switzerland, 2024: 239-254.

[5] Tang Y, Zhang J, Qin X, et al. Reason-before-retrieve: One-stage reflective chain-of-thoughts for training-free zero-shot composed image retrieval[C]//Proceedings of the Computer Vision and Pattern Recognition Conference. 2025: 14400-14410.

[6] Yang Z, Xue D, Qian S, et al. Ldre: Llm-based divergent reasoning and ensemble for zero-shot composed image retrieval[C]//Proceedings of the 47th International ACM SIGIR conference on research and development in information retrieval. 2024: 80-90.

**Questions:**

1.	Why not compare ImageScope in your introduction and related works?

2.	Why not acknowledge OSrCIR’s reflective CoT explicitly in the paper?

3.	Why not compare LDRE with your Retrieval-Driven Caption Optimization module?

4.	What is the inference cost (e.g., average API calls, latency, compute requirements) relative to baselines, and clarify how efficiency was measured in your comparison tables?

5.	What is the performance of different query fusion methods, rather than Slerp?

6.	What is the influence of different modules of their CoT method?

7.	What is the performance of different kinds of MLLM (i.e., GPT, LLaVA)?

8.	What is the performance on CLIP-G?

---

> ### Author Response · Authors · 2025-11-19
> **Rebuttal for ZGRB**
>
> # Q1: Reflection on intermediate retrievals has been explored in prior work
> For [1], regarding CIR tasks, they did not perform query expansion based on retrieval feedback. For [2], they focused on text-image scenarios and required genuine human feedback signals to annotate which retrieval results were relevant. For [3], user feedback was similarly required. These are essentially multi-turn interactive retrieval tasks and bear no relation to ZS-CIR. We are the first ZS-CIR framework to perform self-reflection on results.
>
>
> # Q2: the Historical Query Fusion module appears to overlap with Slerp-TAT
> The Slerp function is not an original invention of Slerp-TAT; we merely employ it to iteratively update the query vector. Slerp-TAT exists to fuse textual and image features. Our objectives differ, and we present performance comparisons of various fusion strategies in Appendix A.4, where Slerp emerges as the optimal approach.
>
>
> # Q3: Limit technology contribution
> OSrCIR does not reflect upon the retrieval results, but merely reflects upon the query. Moreover, the CoT approach is not unique to CIR. Following your line of reasoning, OSrCIR is merely a combination of the CIR task and CoT, lacking any innovation whatsoever.
>
> Our purpose in employing Retrieval-Driven Caption Optimization is to furnish MLLM with richer contextual information, enabling it to implicitly learn the linguistic preferences of the embedding model. It is a fundamentally different approach to LDRE! LDRE employs random sampling to generate captions for images within the query, thereby aiding in synthesising textual queries. We instead perform caption optimisation on the images within the candidate set, providing the optimised captions as context to the MLLM for in-contextual learning. This enables the MLLM to acquire partial prior knowledge of the embedding model.
>
> Slerp is merely an iterative method for updating the query vector and does not constitute our primary contribution. Moreover, slerp itself is not an original invention of Slerp-TAT.
>
> Each step we take differs fundamentally from previous approaches. Not only do our objectives diverge, but even the technical details are distinct. For instance, LDRE requires generating numerous captions for every query, which proves inefficient in practical applications. In contrast, we generate captions for images within the candidate set and employ our ranking strategy to obtain the optimal representation. This process can be fully pre-processed offline, incurring no additional overhead during actual inference. Similarly, our CoT approach is retrieval-driven, fundamentally distinct from methods like OSrCIR which directly reason over queries. Retrieval-Driven Caption Optimisation is inherently aimed at updating the query vector, with slerp representing merely one method of doing so. Appendix A.4 details various update approaches, confirming slerp as the most effective.
>
> # Q4: Potential test-time leakage
> We do not understand your point. All our filtering is conducted via MLLM; why would there be a risk of disclosure? An α=0.8 indicates that the original query's overall semantic meaning is largely sound, but errors occur in the finer details. By fusing these through slerp, we can correct such errors to retrieve accurate results. We see no issue with this approach.
>
>
> # Q5: Heavy pipeline and questionable efficiency comparisons
> We reiterate that we solely generate captions for images within the candidate pool, a process that can be pre-processed offline. This is because the candidate retrieval pool remains virtually unchanged in practice. During inference, we merely invoke the MLLM multiple times with no additional overhead! In this respect, we are significantly more efficient than LDRE, which must generate numerous captions for images within the query in real time.
>
> # Q6: Insufficient implementary details
> We apologise for the lack of clarity in our presentation. We will provide pseudo-code in the camera-ready version for better understanding.
>
>
> # Q7: Concerns about the hallucination issues
> We present a qualitative analysis of failures in Appendix A.6. While errors do occur due to certain illusory factors, we consider this acceptable given the enhanced performance achieved.
>
> # Q8: Insufficient ablation studies
> We will supplement further ablation in camera-ready, and we have already presented the effects of other fusion methods in Appendix A.4.

---

### Official Review · Reviewer_9D9t · 2025-10-31

**Soundness:** 3
**Presentation:** 1
**Contribution:** 2
**Rating:** 2
**Confidence:** 5

**Summary:**

This paper proposes CoRR, a training-free framework for zero-shot composed image retrieval (ZS-CIR). Unlike existing methods that generate queries in a single step, CoRR introduces an iterative "retrieval-reflection-refinement" loop. It uses a Multimodal Large Language Model (MLLM) to analyze initial retrieval results, identify discrepancies with the user's intent through Chain-of-Thought reasoning, and then refine the textual query. To ensure stability, it incorporates a spherical linear interpolation (Slerp) technique for fusing historical queries and a retrieval-driven strategy to optimize captions for the embedding model. Evaluated on multiple benchmarks, CoRR significantly improves retrieval accuracy over state-of-the-art baselines without requiring any task-specific training.

**Strengths:**

1. This paper introduces an iterative "retrieval-reflection-refinement" loop concept, which can improve retrieval accuracy to some extent.
2. The paper is comprehensive in its experimental design, particularly in the ablation studies that examine the impact of various parameters.
3. The proposed method is a plug-and-play module compatible with different approaches, demonstrating its broad applicability.

**Weaknesses:**

1. Line 152 states that "ZS-CIR methods avoid this by converting visual features to textual representations for retrieval," but not all ZS-CIR methods adopt this Image-to-Text (I2T) approach.

2. Figure 1(a) depicts previous methods as fully utilizing MLLMs for retrieval. Firstly, this characterization is overly narrow, as no effective method is so simplistic or relies entirely on MLLMs for retrieval without other design elements. Secondly, Figure 1(a) itself illustrates an I2T method, which does not represent all ZS-CIR methods.

3. There is an inconsistency between the description in Section 3.2 and Figure 2(a). Section 3.2 describes processing the reference image and modification text directly with the embedding model, whereas Figure 2(a) shows them being processed first by an MLLM.

4. What is the relationship between the "synthetic captions" mentioned in line 194 and the "high-quality captions" mentioned in line 261? Furthermore, the description of the ranking and selection process for these high-quality captions in Section 3.5 is too brief and difficult to understand. For instance, the "I" mentioned in line 263 likely refers to the entire dataset, but this is not explicitly stated, which could cause ambiguity for readers. The explanation of Equation 4 is also not sufficiently clear and straightforward.

5. A critical reference for model comparison is missing and should be included: SEIZE [1].

[1] Semantic Editing Increment Benefits Zero-Shot Composed Image Retrieval. ACM Multimedia 2024: 1245-1254

**Questions:**

1. You should revise Figure 1 and the corresponding descriptions in the paper, as the focus of your paper is on ZS-CIR methods in general, not solely on I2T-based approaches.

2. Which one is correct—the description in Section 3.2 or the shown details in Figure 2(a)?

3. What is the relationship between the "synthetic captions" mentioned in line 194 and the "high-quality captions" mentioned in line 261?

4. The meaning of the coordinate system in Figure 2(c) is difficult to interpret. What do the differently colored points represent?

---

> ### Author Response · Authors · 2025-11-19
> **Rebuttal For 9D9t**
>
> # Q1: Not all ZS-CIR methods adopt this Image-to-Text (I2T) approach
>
> Even approaches such as SEARLE, which belong to the category of Textual Inversion, are essentially concerned with converting images into textual representations. To our knowledge, most ZS-CIR systems convert images into text (whether in the form of captions or embeddings). We believe our approach possesses sufficient generality.
>
>
> # Q2: Depicts previous methods as fully utilizing MLLMs for retrieval
> What we wish to convey is that the previous method merely analysed the query through MLLM, whereas we will conduct our analysis based on retrieval results. The issue you raised is not the core point we intended to express, as we have already elaborated on this in Line 053.
>
> # Q3: Different between Section 3.2 and Figure 2 (a)
> This is a mistake in our figure preparation. The description in Section 3.2 is correct. Figure 2(a) is intended to emphasise that our approach requires iterative retrieval, hence the omission of the query initialisation process in the figure. We shall rectify this in camera-ready version.
>
>
> # Q4: Relationship between the "synthetic captions" and "high-quality captions"
> Synthetic captions represent the optimal captions selected for each image through our "retrieval-driven caption optimisation" process. "High-quality captions" is merely a descriptive term for the captions outlined in BLIP2 and does not refer to any specific entity. To prevent ambiguity, we shall amend this to "captions" in camera-ready.
>
>
>
> # Q5: The explanation of Equation 4 is not sufficiently clear and straightforward
> Our sorting strategy shown in Equation 4, in simple terms, prioritises the highest relevance whenever the correct image can be retrieved via caption. Consequently, we first sort in ascending order by the ranking of the retrieved correct images, then sort in descending order by relevance scores. A more detailed explanation will be provided in camera-ready.
>
>
> # Q6: SEIZE for model comparison is missing
> The SEIZE approach shares similarities with LDRE. We have selected only one for comparison. We will supplement this subsequently.
>
>
> # Q7: The meaning of the coordinate system in Figure 2(c) is difficult to interpret
> Figure 2(c) illustrates the relative positions of different items within the embedding space. The coordinate system is merely employed to indicate that this image represents a visualisation space. Yellow denotes the initial query embedding position, red indicates the target embedding position, while green signifies the query embedding updated via our method. Blue represents the embeddings of other items.

---

### Note · Authors · 2025-12-03

I have read and agree with the venue's withdrawal policy on behalf of myself and my co-authors.